# Do Lateral Views Help Automated Chest X-ray Predictions?

**Hadrien Bertrand**[*1]                                                  HADRIEN.BERTRAND@MILA.QUEBEC

**Mohammad Hashir**[*1,2]                                    MOHAMMAD.HASHIR.KHAN@UMONTREAL.CA

**Joseph Paul Cohen**[1,2]                                          JOSEPH.PAUL.COHEN@MILA.QUEBEC

[1]*Mila, Quebec Artificial Intelligence Institute*
[2]*Université de Montréal*

**Editors:** Under Review for MIDL 2019

## Abstract

Most convolutional neural networks in chest radiology use only the frontal posteroanterior (PA) view to make a prediction. However the lateral view is known to help the diagnosis of certain diseases and conditions. The recently released PadChest dataset contains paired PA and lateral views, allowing us to study for which diseases and conditions the performance of a neural network improves when provided a lateral x-ray view as opposed to a frontal posteroanterior (PA) view. Using a simple DenseNet model, we find that using the lateral view increases the AUC of 8 of the 56 labels in our data and achieves the same performance as the PA view for 21 of the labels. We find that using the PA and lateral views jointly doesn't trivially lead to an increase in performance but suggest further investigation.

**Keywords:** Chest X-ray, classification, multilabel, multi view

## 1. Introduction

Most automated radiology prediction models use only posteroanterior (PA) views to make a prediction (Wang et al., 2017; Rajpurkar et al., 2017; Lakhani & Sundaram, 2017; Cohen et al., 2019) as the PA view is often the only available one in public datasets. In many hospitals, the lateral view is infrequently used and usually replaced by a CT scan, as it is difficult to read without specific training (Feigin, 2010). But a CT scan uses a larger dose of radiation, and is only ordered if the PA view is insufficient to diagnose, adding a latency in the diagnosis and risk to the patient.

However, there are specific cases in which the lateral view provides information for diagnosis that isn't clear or visible on the PA view (Shiraishi et al., 2007; Feigin, 2010; Ittyachen et al., 2017). For example, up to 15% of the lung can be obscured by cardiovascular structures and the diaphragm (Raoof et al., 2012). The question we investigate in this work is whether a neural network can make a better prediction using the lateral view or the posteroanterior view, across a wide variety of diseases and conditions. If so, we can look further into how to best augment models to use both modalities.

The release of PadChest (Bustos et al., 2019), a large-scale public chest X-ray dataset that includes paired PA and lateral views, provides us with the opportunity to give a preliminary answer to this question.

---

[*] Contributed equally

## 2. Data and methods

We use the PadChest (Bustos et al., 2019) dataset which is comprised of 160,000 chest X-rays and reports gathered from a Spanish hospital spanning over 67,000 patients with multiple visits and views available. The images have been annotated with various types of radiological findings and differential diagnoses, with 27% of the annotations created manually by physicians and the rest extracted from the report by a recurrent neural network.

For our analysis, we extract a single visit from only those patients who have both PA and lateral views available resulting in a total of 30,699 patients. We resize the images to $224 \times 224$ pixels, utilizing a center crop if the aspect ratio is uneven, and scale the pixel values to $[-1, 1]$ for the training. Each visit can have any number of labels from a total of 194. Since the PadChest dataset defines a hierarchy of labels, we mapped the labels to their respective top level one, in order to maximize the number of images for each label. From those top level labels, we retain only those that occur in at least a 100 patients and combine the rest into "other" resulting in 56 total labels. Some of them are of low clinical interest, such as "electrical device" or "artificial heart valve", however they provide a sanity check on the results of the models.

The model we use is a DenseNet (Huang et al., 2017). This is a convolutional neural networks defined in blocks. Each block contains a set of convolutional layers, where the input of a layer is the concatenation of the output of every preceding layers in the block, making the network densely connected. In between blocks are pooling layers. At the end, there is a linear layer with as many units as we have labels, followed by a sigmoid.

## 3. Experiments

We trained two DenseNets: one on only PA images and the other on only lateral images with a 60-20-20 split between our training, validation and test sets. We ran all models 5 times with different seeds for the random data splits and model initialization for 40 epochs with a batch size of 8 and a learning rate of 0.0001. All models are trained with the Adam optimizer with a binary cross-entropy loss that is weighted for each label according to their frequency. The class weights are applied only to the positive examples and were computed by dividing the total number of samples in the particular split by the number of samples in the class. As this led to weights ranging from 1 to 250 for the rarest labels, we then multiplied them by 0.1, and clamped the resultant value in $[1, 5]$. The code for extracting the data and training the models is publicly available on GitHub.

For testing, we load the model with the weights from the epoch where it achieved the highest area under the ROC curve (AUC) on the validation set. We visualize the results on the test set in Figure 1. For 26 labels, the PA view was more informative. For 8 labels, it was the lateral view, and for the 21 remaining labels both views where similarly informative. There is a high variance for some of the labels, as shown by the error bars, suggesting the need for further testing.

Concerning the absolute performance, the average weighted AUC is 0.79. This is encouraging as to the quality of the dataset, since the model we used was simple and we used only a subset of the available images.

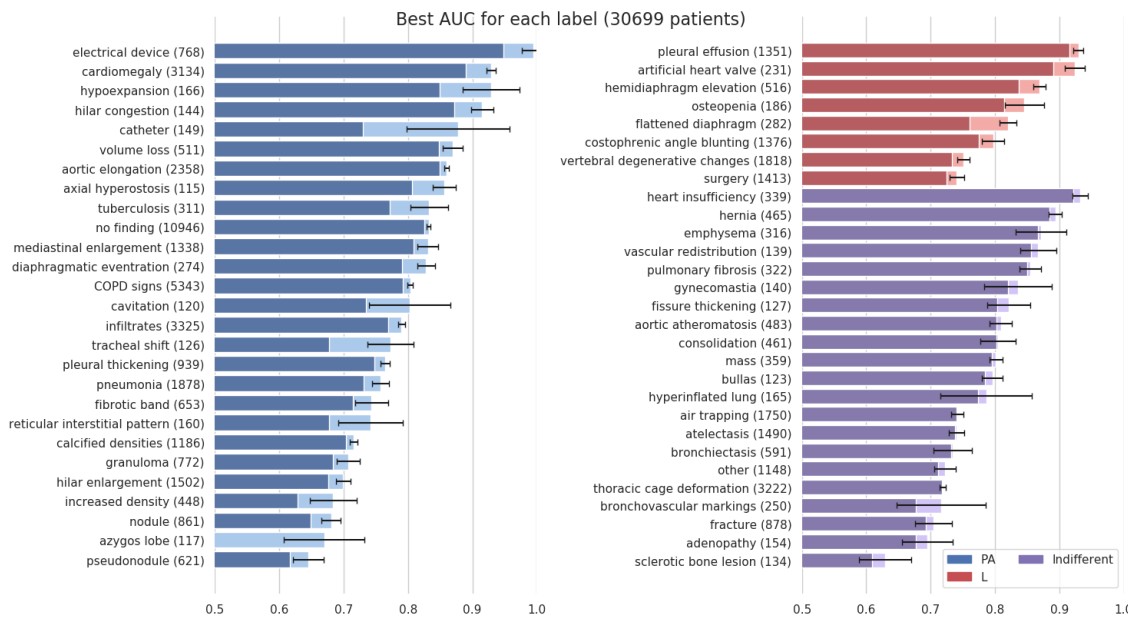

Figure 1: AUC of the best model for each label. (Blue) PA was better compared to L. (Red) L was better compared to PA. (Purple) Both networks had performance difference inferior to the standard deviation across seeds.

## 4. Conclusion

We trained a model on either PA or lateral images, and found that the lateral view performs better for 8 labels, namely pleural effusion, artificial heart valve, hemidiaphragm elevation, osteopenia, flattened diaphragm, costophrenic angle blunting, vertebral degenerative changes and surgery. This suggests that using the lateral images can help for certain prediction tasks, though a more extensive validation is required.

A natural question to ask is if combining both views would improve further the results. There are different ways to do this combination such as stacking the views on the input channels or using a model like DualNet (Rubin et al., 2018) or HeMIS (Havaei et al., 2016) that process each view separately before combining them. Testing those methods, we found that they give an increased AUC for some labels for any given split of the data, but aggregating the results across splits shows a high variance for individual labels, and an overall lower performance than the PA or L only models. While this suggests there is value in using both views jointly, finding a robust way to do so is non-trivial and require further investigation.

There are also limitations from the PadChest dataset. Most labels where extracted from reports by a RNN, making them partly unreliable. There is a bias in the data, as the images come from a single hospitals. Both points can be addressed by validating the results on other datasets such as MIMIC-CXR (Johnson et al., 2019) and CheXpert (Irvin et al., 2019).

## Acknowledgments

This work is partially funded by a grant from the Fonds de Recherche en Sante du Quebec and the Institut de valorisation des donnees (IVADO). This work utilized the supercomputing facilities managed by Mila, NSERC, Compute Canada, and Calcul Quebec. We also thank NVIDIA for donating a DGX-1 computer used in this work.

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
