# OpenReview forum: "Do Lateral Views Help Automated Chest X-ray Predictions?"
_MIDL.io/2019/Conference/Abstract — MIDL Abstract 2019_

### Official Review · AnonReviewer2 · 2019-04-24
**good analysis with existing methods on new data**

**Rating:** 3
**Confidence:** 3

**Review:**

This abstract evaluated two DenseNets, focusing each on separate projections, for diagnosis of chest X-Rays.
The abstract does not introduce any methodological novelty but is well written and executed. The work shows performance on a new chest X-Ray dataset from Spain: PadChest for various diagnosis.
The question that is asked in the title is not clearly answered but sufficiently well discussed.

Re the dataset, "images have been annotated with various types of radiological findings"
it might be beneficial to (re)state earlier if PadChest uses the same error-prone ways of automatic information extraction from clinical reports as the CheXNet data or if there are any differences in this process.

---

### Official Review · AnonReviewer1 · 2019-04-25
**empirical study, interesting for a narrow audience**

**Rating:** 2
**Confidence:** 3

**Review:**

This abstract presents an empirical study, testing whether the use of a less frequent lateral radiograph may help image classification of chest X-ray, where posterior-anterior (PA) images are much more frequent. The results are interesting, but limited to a narrow audience (chest X-ray, or a handful of other imaging problems that face the same dilemma, e.g., mammography). I missed further experiments attempting to combine the two views, even though I understand the space constraints.

---

### Decision · Program_Chairs · 2019-05-06
**Acceptance Decision**

Accept